# Elucidating the mechanism of heterogeneous Wacker oxidation over Pd-Cu/zeolite Y by transient XAS

Jerick Imbao [1,2], Jeroen A. van Bokhoven[1,2✉], Adam Clark [1] & Maarten Nachtegaal [1✉]

The heterogenization of Wacker catalysts using chloride-free systems can potentially be a good alternative for the commercial homogeneous Wacker oxidation of ethylene, which utilizes excessive aqueous chloride solvents. However, the mechanism of the heterogeneous system has not been clarified, preventing the rational design of better catalysts. Here, we report a transient X-ray absorption spectroscopic (XAS) investigation of the heterogeneous Wacker oxidation over Pd-Cu/zeolite Y coupled with kinetic studies and chemometric analysis. Insight is obtained by operando quickXAS allowing the quantitative determination of rates and thereby revealing a rapid redox reaction involving copper. Our work demonstrates that copper is not only the site of oxygen activation, but is also involved in the formation of undesired carbon dioxide. Without detecting the presence of Cu(0) and Pd(I), our results suggest that two one-electron transfers to two Cu(II) ions to reoxidize Pd(0) is at work in this heterogeneous Wacker catalyst.

[1] Paul Scherrer Institute, CH-5232 Villigen, Switzerland. [2] ETH Zurich, Institute for Chemical and Bioengineering, CH-8093 Zurich, Switzerland. ✉email: jeroen.vanbokhoven@chem.ethz.ch; maarten.nachtegaal@psi.ch

Since its discovery, the liquid-phase commercial homogeneous Wacker oxidation of ethylene has been the primary source of acetaldehyde worldwide. Although the development of Cativa and Monsanto processes resulted in a decrease of acetaldehyde demand for acetic acid production, the global market of acetaldehyde is still projected to rise and exceed 1.4 million tons by 2022 due to the steady growth in its consumption for the production of pyridine and pyridine bases for agricultural chemicals; pentaerythritol and acetate esters for surface coatings; and butadiene for synthetic rubbers and polymers[1–7].

However, the corrosion and formation of chlorinated byproducts from the excessive use of aqueous chloride solvents place hefty ecological constraints on this process. Consequently, heterogenization of chloride-free Wacker catalyst systems had been attempted to overcome these problems. Better isolation of products can be achieved by preparing the catalysts on different supports including palladium- and copper-exchanged zeolites[8–15], palladium clusters on titania[16], and palladium on carbon nanotubes[17]. Other chloride-free systems have been proposed, such as palladium deposited on vanadium pentoxide[18–20], palladium on vanadia nanotubes[21] and palladium and manganese salts of heteropoly acids on silica[22] with different co-catalysts replacing the Cu(II)/Cu(I) redox couple. However, most of these catalyst systems either lacked sufficient activity or stability.

Pd-Cu/zeolite Y is one of the most active and stable heterogeneous Wacker catalysts reported in the literature but its mechanism is not understood. One of the few attempts to propose a mechanism for the heterogeneous Wacker process over Pd-Cu/zeolite Y was reported more than 20 years ago by Espeel and coworkers, suggesting that it is similar to the homogeneous PdCl$_2$–CuCl$_2$ system[10,11]. They hypothesized a trinuclear Cu(II)-OH-Pd(II)(NH$_3$)$_2$-OH-Cu(II) active site, which was difficult to identify directly spectroscopically due to the technological limitations to study reaction intermediates in fast catalytic processes at the time.

Synchrotron-based X-ray absorption spectroscopy (XAS) has developed to be the method of choice to determine redox-type reaction mechanisms by studying the electronic and local atomic structures of the metal of interest within the heterogeneous catalysts under operating conditions[23,24]. However, there are no reported XAS studies focused on deciphering the active sites in heterogeneous or homogeneous Wacker oxidation. Traditionally, the acquisition of a single XAS spectrum takes tens of minutes preventing the investigation of the rate-limiting steps and active sites of fast processes. The advent of high brilliance synchrotron sources, together with instrument developments, allows now to monitor structural changes occurring on sub-second to minute timescales, which would not have been possible a decade ago[25,26].

The complexities in the interactions of the palladium and copper centers with the reactants and short life-times of intermediates during the Wacker process make it difficult to unambiguously identify the active species and the role, or lack thereof, of each component in the reaction mechanism under steady-state conditions. Therefore, we used transient XAS, with which the dynamic speciation of palladium and copper can be determined. When the kinetically relevant step is directly affected by disturbing the system, be it by changing the pressure, concentration or temperature, the active site yields a quick response while the spectator species would elicit a slower response[24].

In this work, we investigate the mechanism of the heterogeneous Wacker oxidation of ethylene to acetaldehyde over Pd-Cu/zeolite Y through the synergistic combination of kinetic, spectroscopic and chemometric studies. Kinetic studies reveal the variable order in oxygen at different partial pressures. Transient XAS at these different oxygen reaction orders shows a rapid redox reaction involving the Cu(II)/Cu(I) couple, demonstrating

directly for the first time that oxygen is activated on copper in the heterogeneous Wacker system. The rate-limiting step is identified as Cu(I) reoxidation at low oxygen partial pressure and Cu(II) reduction at high oxygen partial pressure. Concomitant fast changes in the rate of formation of carbon dioxide by-product show that Cu(II) is also involved in the total oxidation of ethylene. Finally, the slow Pd(0) formation leads to loss of Wacker activity by lowering the number of active Pd(II), which is partially reactivated via the gradual reoxidation by Cu(II).

## Results and Discussion

**Kinetics of the Wacker reaction**. The materials utilized in this work were prepared by aqueous ion exchange of the sodium form of zeolite Y and denoted as Pd(*a*)Cu(*b*)-Y [*a,b*=wt%; compositional (Supplementary Table 1) and physicochemical characterization (Supplementary Figs. 1, 2; Supplementary Table 2)]. High-angle annular dark field-scanning transmission electron microscopy (HAADF-STEM) and elemental mapping of Pd1Cu5-Y using energy-dispersive X-ray (EDX) spectroscopy (Supplementary Fig. 1) show that palladium and copper are homogeneously distributed in the zeolite after ion-exchange. Furthermore, the temperature-programmed reduction (TPR) profile of Pd1Cu5-Y (Supplementary Fig. 2) exhibits a single-peak feature indicating that their redox properties are coupled and that palladium and copper interact with each other. Hence, these results suggest that palladium and copper are in close proximity in Pd1Cu5-Y. Testing zeolites containing either or both palladium and copper under Wacker conditions (Supplementary Fig. 3) showed that only copper is inactive and that palladium is the active site for ethylene oxidation. Without copper, there was little activity towards ethylene oxidation over Pd2-Y, albeit catalytic. Clearly, the synergistic presence of both copper and palladium resulted in a significantly higher Wacker activity over Pd1Cu5-Y. Thus, we studied the kinetics of the optimized catalyst, Pd1Cu5-Y[27], for the Wacker oxidation of ethylene under low conversions (<10%) to assume differential reactor conditions, which ensures a minimal concentration gradient from entrance to exit of the catalyst bed.

Figure 1 depicts the dependence of ethylene oxidation on the partial pressures of water, ethylene and oxygen. The apparent reaction orders with respect to water, ethylene and oxygen are listed in Table 1 and are estimated from the linear slope of the double-logarithmic plot of the rate of acetaldehyde formation as a function of partial pressure. At constant oxygen and ethylene partial pressure, the concentration of water in the gas feed has a positive influence on the reaction rate to ca. first-order (0.7) extent (Fig. 1a). The reaction order of water in the homogeneous system could not be measured because it is utilized in large excess as solvent[28]. In the heterogeneous oxidation of ethylene, previous studies have reported a first-order dependence on water[11]. Figure 1b shows that the apparent reaction order with respect to ethylene is 0.7, which is close to the first-order dependence observed in both homogeneous[28–30] and heterogeneous Wacker oxidation[11].

Varying the oxygen partial pressure leads to changes in the apparent reaction order of oxygen (Fig. 1c). At high oxygen partial pressures (0.5–5 kPa), the apparent order is −0.05, close to zero, indicating high oxygen coverage on the catalyst, which is similar to the homogeneous system[28]. Conversely, the apparent reaction order of oxygen becomes roughly half (0.46) at lower partial pressures (0.01–0.5 kPa), suggesting the dissociative adsorption of oxygen onto the active sites that have low oxygen coverage. These findings signify that at high oxygen partial pressure, the catalyst surface is saturated with oxygen and the reaction rate depends on the gas-phase concentration of water and ethylene. At low oxygen partial pressure, the number of

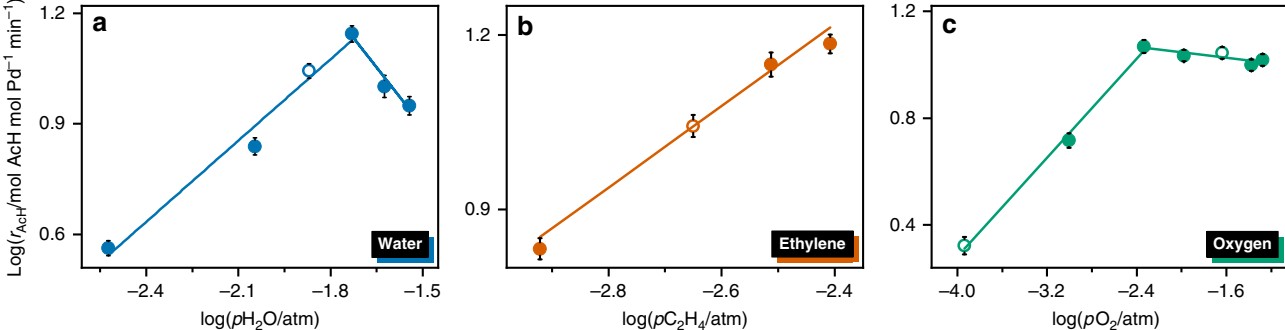

**Fig. 1 Kinetics of Wacker oxidation over Pd1Cu5-Y.** Double-logarithmic plots of the dependence of the rate of acetaldehyde ($r_{AcH}$) formation on varying reactant partial pressures of **a** water, **b** ethylene, and **c** oxygen over Pd1Cu5-Y. The partial pressure of helium was adjusted so that the contact time does not vary. Hollow points in **a** and **b** correspond to the partial pressure values used in the standard Wacker conditions. Hollow points in **c** show the partial pressure values used in the transient measurements done in time-resolved XAS. The error bars in **a–c** indicate the standard deviation ($n = 3$ independent samples).

**Table 1 Comparison of homogeneously and heterogeneously catalyzed Wacker oxidation.**

| Order | Heterogeneous[a] | Homogeneous[b] |
|---|---|---|
| Water | 0.73 (±0.05) | Not measured |
| Ethylene | 0.70 (±0.10) | 1 |
| Oxygen | −0.05 (±0.02) → 0.46 (±0.03) | 0 |

[a]Apparent reaction orders determined from this study: Pd1Cu5-Y at 378 K, 1 atm.
[b]From refs. [28–30].

active oxygen species diminishes and the gas-phase concentration of oxygen has a positive influence (half-order) on the reaction rate. Based on the results of these kinetic studies, transient XAS experiments were designed to monitor structural changes in the catalyst in each of the kinetic regimes by varying the oxygen partial pressure.

**Cu and Pd K-edge XAS**. Figure 2 shows the time-resolved Cu and Pd K-edge X-ray absorption near-edge structure (XANES) spectra of Pd1Cu5-Y before and during the transient experiments. The structural parameters derived from the curve fitting of the Cu and Pd K-edge extended X-ray absorption fine structure (EXAFS) spectra (Supplementary Fig. 4) are listed in Table 2. The corresponding Fourier transformed (FT) EXAFS spectra are presented in Fig. 3. The presence of Cu(II) species in the as-prepared Pd1Cu5-Y is identified by a pre-edge feature at 8977.5 eV due to 1s→3d transition[31,32] (Fig. 2a). Moreover, the spectrum of the as-prepared catalyst resembles that of the hexaaquacopper(II) ion reference (Supplementary Fig. 11) indicating the presence of mobile hydrated copper complexes having an intense white line peak at 8996 eV, which diminished after pre-treatment in oxygen at 378 K due to partial dehydration. Accordingly, the Cu K-edge FT-EXAFS spectra of the as-prepared and pre-treated catalyst show only one major peak associated with oxygen neighbors at a distance of 1.94 Å, suggesting the presence of isolated Cu(II) species and thus no chemical bonds between copper and palladium.

The Pd K-edge XANES spectrum of the as-prepared Pd1Cu5-Y is closely similar to that of the tetraamminepalladium(II) precursor (Supplementary Fig. 12), which is indicative of the presence of Pd(NH$_3$)$_4^{2+}$ ions after ion-exchange. This is further supported by the ATR-IR spectrum of Pd1Cu5-Y (Supplementary Fig. 5) exhibiting a weak band at 1310 cm$^{-1}$ attributed to symmetric deformation of NH$_3$ coordinated to Pd(II)[10,11].

Moreover, the FT-EXAFS spectrum of the as-prepared catalyst shows a lone significant backscattering contribution between 1–2 Å due to light elements like oxygen and nitrogen in the first coordination shell. The FT-EXAFS spectrum is featureless at higher distances suggesting the absence of particulate palladium oxide and directly coordinating copper atoms in the as-prepared catalyst. Since the backscattering amplitudes of oxygen and nitrogen are nearly identical, it is not possible to distinguish them by EXAFS analysis. Thus, the EXAFS spectrum of the as-prepared catalyst was fitted with a single Pd-O/N scattering path. From the best fit parameters summarized in Table 2, the palladium ions are found to be coordinated to 3–4 O/N atoms at 2.01 Å. The pre-treatment of Pd1Cu5-Y in oxygen at 378 K did not result in any significant change as observed in its Pd K-edge XANES and EXAFS spectra. EXAFS analyses do not provide evidence for the direct coordination between copper and palladium in Pd1Cu5-Y. However, its XANES spectra show the presence of solvated copper(II) and palladium(II) complexes in the wide-pored zeolite Y. As previously discussed through the results from STEM-EDX mapping and TPR studies, the close contact and electron transfer between the solvated copper and palladium complexes hint at their mobility within the zeolite, which have been suggested to have similar ionic characters as those of palladium and copper ions in aqueous solutions[8,15].

After pre-treatment in oxygen at 378 K, Pd1Cu5-Y was equilibrated under Wacker conditions for 4 h and the corresponding Cu K-edge XANES spectrum of the catalyst (Fig. 2a) reveals the formation of a minor peak at 8983 eV, which is associated with the formation of Cu(I)[31,32]. The Pd K-edge XANES spectrum of Wacker-equilibrated Pd1Cu5-Y shows that the absorption edge energy at half-height shifted to lower energy, signifying the reduction of a fraction of Pd(II) into Pd(0). From EXAFS analysis, the average Pd-O/N coordination number decreased from 3.5 to 3.1 while the bond distance remained at 2.01 Å. A second Pd-Pd (metallic) coordination shell appeared at 2.76 Å with a coordination number of 1.5, suggesting the reduction of a portion of Pd(II) to metallic nanoparticles[33], which could be linked to the loss of Wacker activity observed in Pd1Cu5-Y (Supplementary Fig. 3). The relatively large pseudo Debye-Waller factor of the metallic Pd-Pd shell suggests a large disorder due to the presence of Pd clusters: increasing Debye-Waller factors is associated with decreasing particle size[34].

After equilibration, the oxygen partial pressure was lowered from 2 kPa to 0.01 kPa to follow the changes in the local geometric and electronic environment of palladium and copper atoms from the zero-order to half-order kinetic regime through the simultaneous acquisition of quickXAS[35] spectra (operando

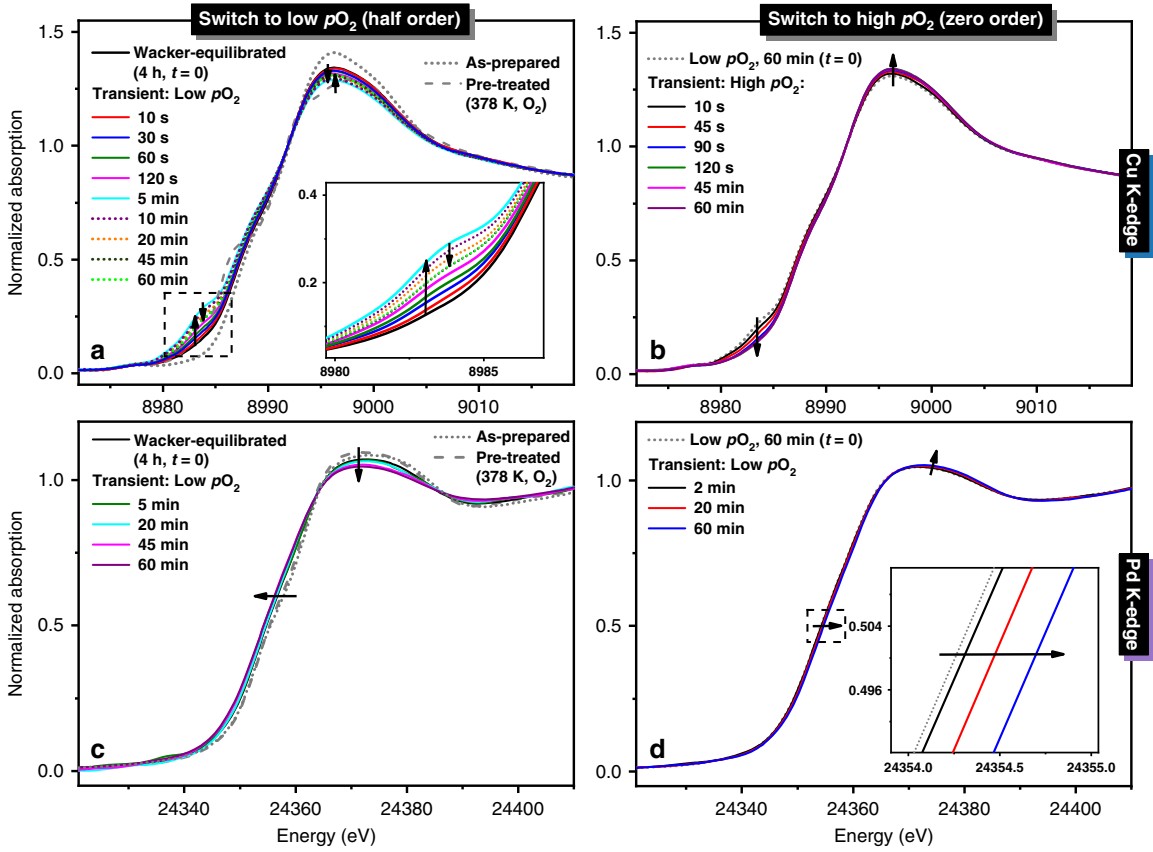

**Fig. 2 Time-resolved Cu and Pd K-edge XANES spectra.** Normalized time-resolved **a**, **b** Cu and **c**, **d** Pd K-edge XANES spectra of Pd1Cu5-Y before (as-prepared, pre-treated, Wacker-equilibrated) and during transient experiments with **a**, **c** low $pO_2$ and **b**, **d** high $pO_2$. Inset: magnification of the **a** minor peak at 8983 eV corresponding to Cu(I) formation and **d** shift in absorption edge energy at half-height. Gas feed composition ($C_2H_4$:$H_2O$:$O_2$) at high $pO_2$ (1:7:10) and low $pO_2$ (1:7:0.05) was balanced with helium to achieve a constant contact time in all experiments.

**Table 2 Best-fit EXAFS parameters for Pd1Cu5-Y.**

| | Scattering pair | $N^a$ | $R(Å)^b$ | $\sigma^2(Å^2)^c$ | $\Delta E(eV)^d$ | R-factor$^e$ |
|---|---|---|---|---|---|---|
| **Cu K-edge** | | | | | | |
| As-prepared | Cu-O | 4.5 (0.2) | 1.94 (0.004) | 0.006 (0.001) | 1.0 (0.6) | 0.002 |
| Pre-treated (378 K, $O_2$) | Cu-O | 4.4 (0.2) | 1.94 (0.004) | 0.006 (0.001) | 3.1 (0.9) | 0.004 |
| Wacker-equilibrated (4 hr) | Cu-O | 4.2 (0.2) | 1.95 (0.004) | 0.006 (0.001) | 1.3 (0.5) | 0.001 |
| Low $pO_2$, 60 min | Cu-O | 3.8 (0.1) | 1.94 (0.004) | 0.007 (0.001) | 0.8 (0.6) | 0.002 |
| High $pO_2$, 60 min | Cu-O | 4.2 (0.1) | 1.94 (0.004) | 0.007 (0.001) | 0.8 (0.5) | 0.001 |
| **Pd K-edge** | | | | | | |
| As-prepared | Pd-O/N | 3.6 (0.2) | 2.01 (0.01) | 0.003 (0.001) | 3.1 (1.0) | 0.009 |
| Pre-treated (378 K, $O_2$) | Pd-O/N | 3.5 (0.2) | 2.01 (0.01) | 0.004 (0.001) | 8.5 (1.3) | 0.010 |
| Wacker-equilibrated (4 hr) | Pd-O/N | 3.1 (0.3) | 2.01 (0.01) | 0.004 (0.001) | 2.5 (1.5) | 0.009 |
| | Pd-Pd | 1.5 (0.9) | 2.76 (0.05) | 0.018 (0.003) | 7.9 (2.5) | |
| Low $pO_2$, 60 min | Pd-O/N | 2.6 (0.4) | 2.01 (0.01) | 0.005 (0.003) | 2.9 (1.9) | 0.014 |
| | Pd-Pd | 1.5 (1.0) | 2.78 (0.07) | 0.015 (0.009) | 2.8 (2.0) | |
| High $pO_2$, 60 min | Pd-O/N | 2.9 (0.3) | 2.01 (0.01) | 0.006 (0.003) | 3.0 (2.0) | 0.047 |
| | Pd-Pd | 1.4 (1.0) | 2.78 (0.07) | 0.014 (0.010) | 3.9 (2.2) | |

*EXAFS* extended X-ray absorption fine structure, standard errors in brackets.
$^a$Number of nearest neighbors determined by fixing the amplitude reduction factor ($S_0^2$) values obtained from the fits to Cu ($S_0^2 = 0.82$) and Pd ($S_0^2 = 0.86$) foil reference spectra.
$^b$Interatomic distance.
$^c$Pseudo Debye-Waller factor.
$^d$Shift of the energy threshold.
$^e$R-factor: goodness of fit between experimental and theoretical data.

XAS setup: Supplementary Fig. 6). The oxygen partial pressure was subsequently increased back to 2 kPa to switch from half-order to zero-order kinetic regime, where XAS spectra were collected for up to 45 min after the switch. These switching experiments were repeated two more times, amassing more than

75,000 spectra from the continuous acquisition for both Cu and Pd K-edges, of which 10 scans were averaged to achieve a temporal resolution of 5 s.

Within tens of seconds after lowering the oxygen partial pressure, qualitative inspection of the Cu K-edge XANES spectra

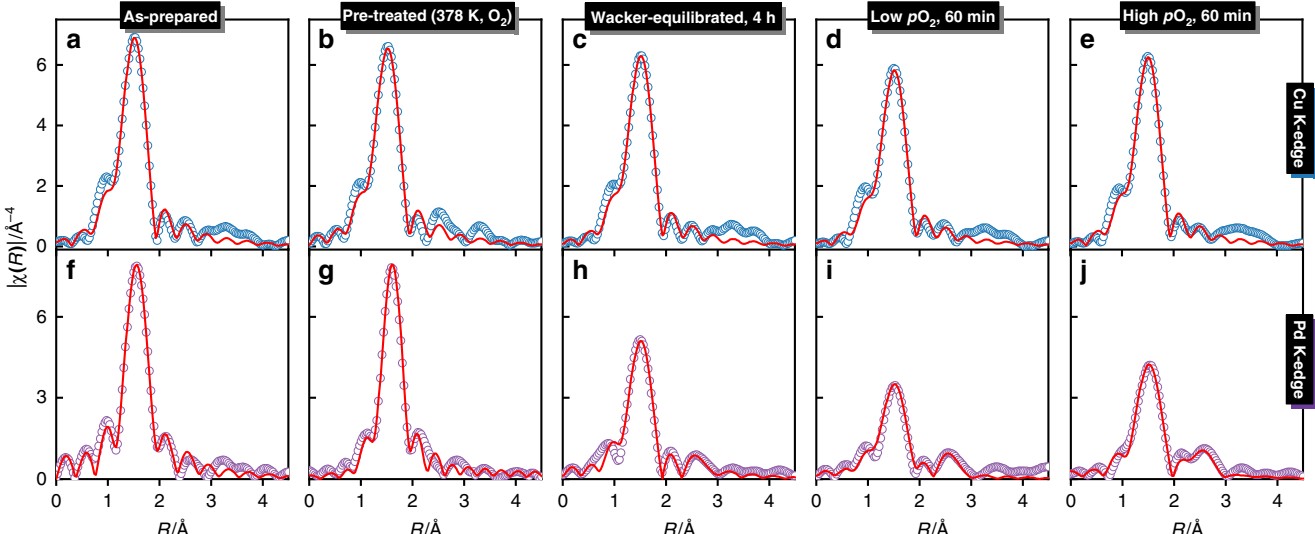

**Fig. 3 In situ Cu and Pd K-edge EXAFS spectra.** The $k^3$-weighted Fourier transform magnitude (non-phase shift corrected) of the **a**–**e** Cu and **f**–**j** Pd K-edge EXAFS spectra of Pd1Cu5-Y before (as-prepared, pre-treated, Wacker-equilibrated) and at the end of each oxygen switching experiment, performed in the 3–11 Å$^{-1}$ range (red traces: best fit). The $k^3$-weighted Cu and Pd K-edge EXAFS spectra are shown in Supplementary Fig. 4.

(Fig. 2a) reveals the immediate diminution of the white line intensity at 8996 eV and formation of a minor peak at 8983 eV, which is associated with the increase in the fraction of Cu(I)[31,32]. After 5 min, the continuum resonance peak at 8996 eV starts to increase slightly while the intensity of the peak at 8983 eV decreases until the 20th minute and barely changed thereafter, suggesting a slow reoxidation of Cu(I) to Cu(II). Figure 2c shows that the Pd K-edge XANES spectra of Pd1Cu5-Y shifted to lower absorption edge energy after lowering the oxygen partial pressure while the intensity of the continuum resonance peak at 24372 eV decreased, albeit more slowly compared with the changes in the Cu K-edge spectra, indicating a gradual reduction of Pd(II) to Pd(0).

Switching the partial pressure of oxygen back to 2 kPa resulted in a further weakening of the peak at 8983 eV in the Cu K-edge XANES spectra with a concomitant increase in the intensity of the white line at 8996 eV (Fig. 2b), which is indicative of Cu(I) reoxidation to Cu(II). The increase in oxygen partial pressure caused a slight shift to higher absorption energy at half-height and an increase in the white line intensity at 24372 eV in the Pd K-edge XANES spectra (Fig. 2d). Furthermore, EXAFS analysis reveals that the Pd-O/N scattering peak intensity (Fig. 3j) and coordination number increased from 2.6 to 2.9, suggesting a slight reoxidation of Pd(0) to Pd(II).

Subsequently, two further repeats of the kinetic regime switching transient experiment were performed (Supplementary Fig. 7). The Cu K-edge XANES spectra of Pd1Cu5-Y after each cycle during all the three switching cycles illustrate the reversibility of the electronic and structural properties of the copper centers. Conversely, some of the palladium centers undergo irreversible changes in the first cycle due to sintering resulting in some extent of deactivation. In the succeeding kinetic switching cycles, changes in palladium speciation became reversible.

**Chemometric analyses of operando Cu and Pd K-edge XANES.** A Multivariate Curve Resolution – Alternating Least Squares (MCR-ALS) chemometric algorithm[36] was employed to disentangle each of the contributions of copper and palladium species directly from experimental data but this requires prior knowledge

of the total number of pure components ($N_{pure}$) characterizing the whole dataset[37,38]. Thus, principal component factor analysis (PCA)[39] was performed to characterize the maximum variance (eigenvalues) of the entire dataset and determine the number of principal components (PC) that are significantly contributing to each spectrum. The results from PCA (Supplementary Fig. 8) and MCR-ALS analyses (Supplementary Fig. 9) identified 3 pure components for copper consisting of Cu(I) and two Cu(II) species (hydrated and dehydrated) while two palladium species, Pd(0) and Pd(II) were assigned to the spectra of the pure components (assignment of spectra by comparisons with experimental reference XANES spectra are shown in Supplementary Figs. 10–12).

The time-resolved speciation of palladium and copper in Pd1Cu5-Y during the transient experiments, derived from the MCR-ALS analysis of their respective XANES spectra, are shown in Fig. 4, together with the corresponding mass spectrometry (MS) response. Due to the dead time (15–30 s) associated with the MS response upon changing the oxygen partial pressure, the time zero in the MS profile was adjusted to when the normalized signal of oxygen ($m/z = 32$) starts to plummet.

At time zero, the decrease in oxygen partial pressure immediately resulted in changes in the MS signals and copper speciation (Fig. 4a, c, e). Within tens of seconds after time zero, a rapid increase in Cu(I) fraction from the reduction of Cu(II) is evident with a concomitant decrease in ethylene conversion. Therefore, the decrease in the rate of copper reoxidation lowers the rate of Wacker oxidation. The quick reduction of Cu(II) to Cu(I) is a direct result of the change in oxygen order from zero to 0.5 and shows that oxygen is activated on copper in the heterogeneous Wacker system.

This reduction of Cu(II) is coupled to the decrease in acetaldehyde formation without significant Pd(0) formation, suggesting that palladium assumes a resting state as Pd(II). Without enough oxygen available, Cu(I) accumulates as its reoxidation becomes slower than the reduction of Cu(II). Hence, reoxidation of Cu(I) to Cu(II) is rate-limiting at low oxygen partial pressure.

Furthermore, the $m/z$ signal for carbon dioxide dropped instantaneously upon lowering the partial pressure of oxygen suggesting that Cu(II) is directly involved in the formation of

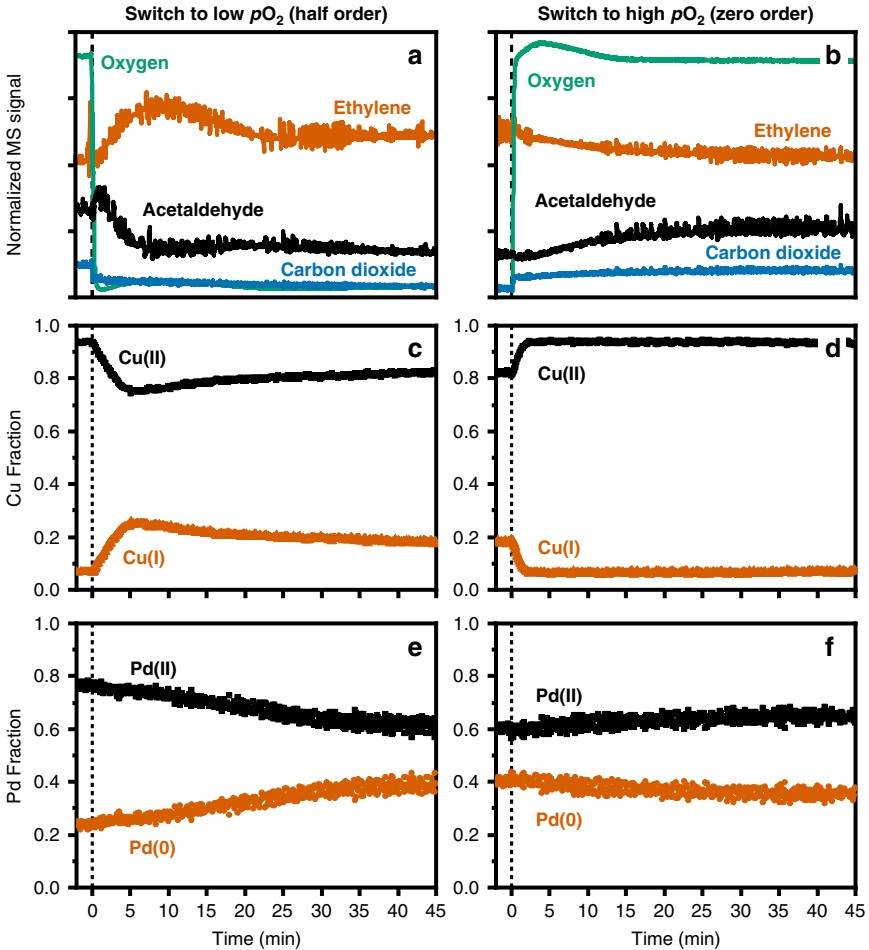

**Fig. 4 Transient oxygen switching experiments. a**, **b** Normalized mass spectrometry (MS) signals of outlet gas components as a function of time during the transient oxygen switching experiments over Pd1Cu5-Y. Corresponding dynamic **c**, **d** copper and **e**, **f** palladium speciation determined from MCR-ALS of time-resolved XANES spectra. The fraction of Cu(II) species is the sum of hydrated and dehydrated copper species, the speciation of which are shown in Supplementary Figs. 13–16. The catalyst was equilibrated under Wacker conditions for 4 h before the first change in oxygen partial pressure ($t < 0$). After changing the partial pressure of oxygen (zero order to half order and back), the catalyst was subjected to the gas feed until the MS signals equilibrated. This kinetic regime switching experiment was carried out two more times and the results are provided in full detail in Supplementary Figs. 13–16. Gas feed composition ($C_2H_4:H_2O:O_2$) at high $pO_2$ (1:7:10) and low $pO_2$ (1:7:0.05) was balanced with helium to achieve a constant contact time in all experiments.

undesired carbon dioxide. Although ethylene is known to adsorb on Cu(I)[40], this interaction is insufficient to lead to full oxidation at low oxygen partial pressures.

Slow reoxidation of Cu(I) to Cu(II) could then be observed after 5 min and loosely followed the gradual formation of Pd(0). The slight reoxidation of Cu(I) and concomitant Pd(II) reduction diminished the number of active Pd(II), leading to catalyst deactivation by Pd(0) formation.

After 60 min of low oxygen partial pressure (Fig. 4b, d, f), increasing the oxygen partial pressure back to the standard conditions led to a quick reoxidation of Cu(I) to Cu(II) within tens of seconds and equilibrated after 120 s, which corroborates the inference that oxygen is activated on copper. A rapid increase in carbon dioxide formation is also evident confirming that Cu(II) is involved in the total oxidation of ethylene, lowering the acetaldehyde selectivity.

However, Cu(I) was not fully reoxidized to Cu(II), suggesting the presence of spectator Cu(I) species that did not participate in the catalytic cycle. This is most likely due to the Cu(I) species that can migrate more easily to the less accessible sites in the zeolite leading to a higher activation barrier for oxidation compared to the active Cu(I) species in the supercages[41].

The fraction of Pd(II) slightly increased exhibiting a slow reoxidation of Pd(0) after switching the partial pressure of oxygen. However, the rapid increase in Cu(II) did not immediately result in an increase in ethylene conversion to acetaldehyde. Formation of acetaldehyde increased more gradually and was coupled to the increase of Pd(II); slow reoxidation of inactive Pd(0) resulted in Pd(II) that again participated in the catalytic cycle.

Moreover, the Wacker activity and the fraction of Pd(II) did not revert to their original values before the start of the transient experiments. Figure 4e shows that 24% of Pd(II) was already reduced to Pd(0) after the 4-hr equilibration under Wacker conditions. The Pd(II) fraction decreased further from *ca*. 76% to 60% after the first switch to half-order kinetics and only increased to 64% at the end of the switch to zero-order kinetics. Relative to the rate before the start of the transient experiments, the rate of acetaldehyde formation decreased to 26% while that of carbon dioxide diminished to 10% after the switch to half-order kinetics. Increasing the oxygen partial pressure resulted in regaining 85% of the original steady-state carbon dioxide formation rate but only 57% of the acetaldehyde formation rate was recovered. These results strongly suggest that the loss of

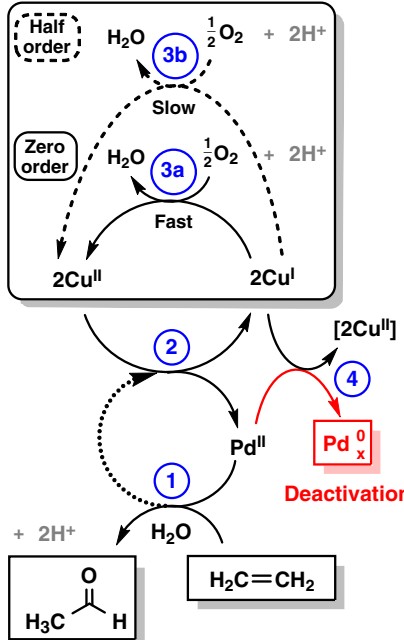

**Fig. 5 Proposed mechanism for heterogeneous Wacker oxidation over Pd-Cu/Y.** The catalytic cycle depicts the results obtained from this study. Reoxidation of palladium (Step 2) is represented by the dotted arrow. The activation of oxygen on copper (boxed) is depicted at high (solid arrow, Step 3a) and low (dashed arrow, Step 3b) oxygen partial pressures. The catalyst deactivation is shown in red.

Wacker activity is a direct result of Pd(II) reduction, which could only be partially reactivated.

In the two subsequent kinetic regime switching cycles (Supplementary Figs. 15, 16; Supplementary Table 3), the rates of acetaldehyde formation and palladium speciation became reversible. This indicates the presence of two different palladium species: Pd(II) (ca. 36%, including those that were already reduced before the start of the transient experiments) that gets irreversibly reduced to Pd(0); and palladium (ca. 4%) that is reversible between Pd(II) and Pd(0) and is most likely situated at the surface of metallic Pd nanoparticles.

Consistent with the qualitative inspection of its XANES spectra, the copper speciation throughout all the three switching cycles show the reversibility of the copper speciation (ca. 10-11%). Thus, the reversible deactivation and reactivation of the catalyst involve redox processes between copper and palladium.

**Mechanism of heterogeneous Wacker oxidation over Pd-Cu/Y.** Based on the findings from this study, a reaction cycle for ethylene oxidation over Pd-Cu/zeolite Y is proposed in Fig. 5.

The pre-treatment in oxygen at 378 K only resulted in the partial dehydration of the copper complexes and did not show any direct coordination between copper and palladium from the EXAFS analysis. However, the XANES spectra of the catalyst revealed that it is composed of mobile hydrated copper(II) and amminated palladium(II) ions. Moreover, $H_2$-TPR studies showed that the redox properties of palladium and copper are coupled while the STEM-EDX images of Pd1Cu5-Y illustrated their homogeneous distribution in the zeolite. These results suggest the interaction and proximity between palladium and copper, which are necessary for efficient electron-transfer.

Step 1 of the reaction mechanism depicts the oxidation of ethylene to acetaldehyde on palladium, which had been demonstrated in

this study and in other previous reports[8–12] as the active site. Reactivation of the catalyst is then presented in Step 2 via Pd(0) reoxidation by neighboring Cu(II). In the absence of copper, a small fraction of Pd(0) could be reoxidized by oxygen as observed through the catalytic acetaldehyde formation over Pd2-Y. However, the lack of observation of Pd(0) intermediates by XAS suggests a very low steady-state concentration. As there is no evidence of Cu(0) and Pd(I) formation, two one-electron transfers to two Cu(II) ions complete the cycle. The activation of oxygen on copper is represented by Steps 3a and 3b (boxed) showing the variable apparent order in oxygen determined from kinetic studies and transient experiments. At low oxygen partial pressures (Step 3b, half-order), Cu(I) reoxidation becomes rate-limiting. Under standard Wacker conditions at high oxygen partial pressure (Step 3a, zero-order), the reoxidation of Cu(I) becomes much faster than the corresponding reduction of Cu(II). Finally, Step 4 shows the slow catalyst deactivation from the reduction of Pd(II) to Pd(0) and concomitant Cu(I) oxidation, which diminishes the number of active Pd(II) species that participate in the Wacker cycle. This is partially reversible upon switching back to the zero-order kinetic regime.

## Methods

**Catalyst preparation and characterization.** Ion-exchanged catalysts were prepared from the sodium forms of zeolite Y ($SiO_2/Al_2O_3$ = 5.1, Zeolyst International). Sodium ions in the zeolite were exchanged with 0.01 M solution of Cu $(NO_3)_2 \cdot 3H_2O$ (>99%, Sigma-Aldrich) at room temperature with a solid/liquid ratio of 1 g/L, mixed continuously for 24 h. Nitrate ions were then removed by washing with deionized water. Subsequently, the remaining sodium ions were exchanged in a 0.1 mM solution of $Pd(NH_3)_4(CH_3COO)_2$ (>99%, Alfa-Aesar) under similar conditions as the previous ion exchange with copper. All the metal-exchanged zeolites were washed with deionized water to remove the acetate ions; filtered; and dried at 373 K overnight. The dried zeolite powder was then pelletized, crushed and sieved to fractions between 100–150 μm. Structural characterization of the ion-exchanged zeolite samples are all described in detail in the Supplementary Methods.

**Catalytic performance test and kinetic studies.** Wacker oxidation of ethylene was carried out in a quartz tube (internal diameter = 6 mm) fixed bed plug flow reactor using CATLAB microreactor (Hiden Analytical) with standard operating conditions of 378 K, 1 atm and space-time (W/$F_0$) of 0.86 kg s mol$^{−1}$ unless stated otherwise. The microreactor is equipped with a furnace and "in-bed" thermocouple for direct measurement of catalyst temperature. The gas feed composition was regulated by mass flow controllers (Bronkhorst EL-FLOW F-201CV). The standard gas feed is composed of 2.3 mol% oxygen, 0.2 mol% ethylene, 1.4 mol% water and 96.1 mol% helium. Water was added to the feed by reacting hydrogen and oxygen using an external homemade reactor bearing a Pt/cordierite catalyst. The gas outlet was analyzed by an online gas chromatograph (MicroGC-MS, SRA) equipped with PoraPLOT U, Molsieve 5 A and StabilWAX columns installed in separate channels connected to a thermal conductivity detector. All the metal-exchanged zeolites were pre-treated in oxygen at 378 K for 1 hr unless otherwise specified.

The kinetics of the reaction was investigated by determining the overall reaction rates and orders with respect to ethylene, water and oxygen at 378 K under low conversions (<10%). When the order of one reactant is tested, its partial pressure was varied while the partial pressures of the other two were fixed. The total flow rate was then adjusted by changing the partial pressure of the inert gas (helium) in order to have a constant contact time in all experiments.

**Time-resolved XAS.** Operando quickXAS measurements were performed in a custom-made quartz capillary reactor cell with quartz wool on both ends of the catalyst to secure the bed (Supplementary Fig. 6). The reactor was heated with two hot air blowers. The controlled gas flows were directed to the reactor and the gas outlet was analyzed by a quadrupole mass spectrometer (Omnistar Pfeiffer). Experiments were performed in steady-state and transient conditions wherein the partial pressure of oxygen is changed from 2 kPa to 0.01 kPa and back to monitor structural and oxidation state changes in the catalyst in different kinetic regimes. Gas feed composition ($C_2H_4:H_2O:O_2$) at high $pO_2$ (1:7:10) and low $pO_2$ (1:7:0.05) was balanced with helium to achieve a constant contact time in all experiments. The sub-stoichiometric amount of oxygen at low partial pressure was chosen from the kinetic studies eliciting the largest changes in the time-resolved XAS spectra. The quickXAS spectra[35] were collected at the SuperXAS beamline of the Swiss Light Source (SLS, Villigen, Switzerland) in transmission mode at the Pd

(24.35 keV) and Cu (8.979 keV) K-edges. The polychromatic beam of the 2.9 Tesla superbend was collimated by a Si-coated (for copper) or a Pt-coated (for palladium) collimating mirror and subsequently monochromatized by a Si(111) channel-cut crystal of the quickXAS monochromator, which oscillated at 1 Hz frequency with an acquisition time of 500 milliseconds per spectrum. The beam was then focused to the sample using a Rh-coated or Pt-coated toroidal mirror to a spot size of $1 \times 0.2$ mm. Details regarding data extraction, processing and analysis are given in the Supplementary Methods.

## Data availability

All data generated and analyzed from this study are archived on the internal servers of Paul Scherrer Institute and are available from the corresponding authors upon reasonable request.

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

## Acknowledgements

We thank the Swiss National Science Foundation (Project 159555) for the financial support. The allocation of beamtime at SuperXAS beamline at the Swiss Light Source is appreciatively acknowledged. We are also grateful to J. Kuzmenko, A. Knorpp, M. Sommerhalder, P. Šot, R. Kopelent, J. Xto and T. Fovanna for their assistance with data acquisition during beamtime. Finally, we thank L. Bani, U. Vogelsang and S. Hitz for all the technical support.

## Author contributions

J.I. prepared the catalysts; designed experiments; and performed catalytic tests, structural characterizations, kinetic studies, transient XAS measurements, PCA and MCR-ALS analyses of the XAS data. A.C. developed the python-based program for processing the quickXAS data. M.N. and J.v.B. designed experiments, analyzed the data and led the project. J.I. wrote the paper. All authors discussed the results and commented on the manuscript.

## Competing Interests

The authors declare no competing interests.

**Additional information**

