## [Peer Review File · Nature Communications]

Reviewers' comments:

Reviewer #1 (Remarks to the Author):

Heterogenization of homogeneous Wacker Pd-Cu catalytic species may be an interesting subject, but the catalysis performance of the Pt-Cu/Y zeolite prepared in this study is low and the detailed characterization of this sort of catalytic materials would be almost meaningless in view of practical applications. The conclusion on the behaviors and catalytic aspects of Pd(II)/Pd(0) and Cu(II)/Cu(I) is not new. The transient XAFS is also conventional, where the XAFS data for Pd and Cu have not been measured simultaneously but independently, and their Pt and Cu data appear to be combined on Figures. The discussion and conclusion on the local arrangements/structures of Cu(II) ions and Pd(II) ions argued from the XANES spectra are not convincing, where the changes in the XANES spectra for Cu and Pd species during the transient processes are very small though the authors made efforts to analyze by the MCR-ALS chemometric algorithm. Other appropriate spectroscopic data for the coordination ligands should be presented for evidence. Further, the responses of Cu(II)/Cu(I) and Pd(II)/Pd(I) in the transient processes of decreasing and increasing O₂ pressure were observed partially and the catalytic system always involved inactive Cu(II) and Pd(0) species heterogeneously in the catalyst. Thus, the current transient XAFS technique may not observe the real kinetic behavior of the active sites. In Figure 4 the quantity of the change in the acetaldehyde response does not agree with those of the changes in the Cu(II)/Cu(I) and Pd(II)/Pd(0) responses. The authors state that Figure 2 and Figure 4 are the transient results, but these are averaged aspects over 5 sec time resolution, and Figure 4 does not show elementary steps in Figure 5 for the catalysis. The authors describe the word “rapid”, “immediate”, etc. in several places, but the time scales are 5 min or so. These slow responses may bring about the low active performance of the catalyst. In the Mechanism section, one-electron transfers from Pd(0) to two Cu(II) ions, but the transient XAFS technique could not observe close locations to each other and no data for the synergistic effect of Cu and Pd on the Wacker catalysis were presented though the authors indicate their synergistic effect on the Wacker catalysis in Supplementary Figure 2. Finally, a proposed mechanism is shown in Figure 5, where unfortunately, the Pd sites as the Wacker reaction site have not been characterized and the argued mechanism is qualitative and even speculative with Cu sites as mentioned above. By careful overview and due to overall inconsistency of the approach and discussion, this manuscript has not provided new insight into the catalysis field, catalyst development and X-ray spectroscopy, and the reviewer cannot recommend this article for publication in Nature Communications.

Reviewer #2 (Remarks to the Author):

The paper 'Elucidating the mechanism of heterogeneous Wacker oxidation over Pd-Cu/zeolite Y by transient XAS' by Nachtegaal et al. describes the changes in Pd and Cu speciation, through operando XAS, as the oxygen partial pressure is varied for the heterogeneously catalysed Wacker oxidation. Overall, the paper is a well-executed piece of work of which the authors have clearly carried out with a great deal of care and rigour. This current paper focuses on the use of XAS to probe the mechanistic steps involved and corroborates many of the ideas put forward in the earlier work by Jacobs (J. Phys. Chem. 1994, 98, 11588-11596) performed using FTIR and EPR. However, comparing the manuscript to the study by Jacobs I am a little unclear regarding what the major breakthrough is in this piece of work? The mechanism proposed has many similarities to the one put forward in that study and I am still unsure whether the data in the current manuscript wholly supports the proposed reaction cycle.

To summarise, I think this is a very good study but the authors need to provide greater justification of the unique insight of this work.

My specific comments on the manuscript are as follows:

- 1) The comments on the advancement of synchrotron sources with respect to time resolution (page 4 lines 7-9) is correct in that things have improved with time – especially since the paper by Jacobs. However, measurements of this type have not been made possible by any recent advancement in the last 10 years. The authors may want to add a greater nuance to this part of the introduction.
- 2) The catalyst is prepared using a consecutive ion exchange process – 1st substituting with Cu and then Pd. The Cu substitution includes the details about the exchange process (concentration of solution, ratio to mass of zeolite, etc.), however, the same details are not given for the Pd step. It would be helpful if this can be included in the methods section alongside providing a justification for the sequential addition of metals and a comment on the total number of ion exchange sites (Na⁺) present per gram of zeolite. How does the PdCu loading introduced compare to the number of ion exchange sites?
- 3) Page 7 lines 22 & 23 – The comment about electron transfer/ion mobility is unsubstantiated and needs further justification.

4) In general the figure captions do not contain sufficient detail, e.g., it would be helpful if the caption for figure 2 could include the ratio of all reactive gases (O₂, H₂O, C₂H₄) present in the feed. This is not really expressed clearly in the manuscript and instead needs to be extracted from the data points highlighted in figure 1. This is quite a useful piece of information to understand the manuscript and should also be mentioned in the main text before the figure is introduced.

5) The changes in the Pd K edge operando XANES study both qualitatively and through the multivariate analysis indicate the presence of Pd(0) increases between the 4h equilibrated form of the catalyst and at 1 h after switching to the low partial pressure of oxygen. There is no reason to suggest that this assessment is incorrect, however, the EXAFS does not show an increase in the Pd-Pd coordination number. It would be of value to comment on this in the manuscript alongside the values for the pseudo Debye-Waller factor. A value of 0.018 is larger than I would expect for 1st shell metallic Pd-Pd and it would be helpful for the authors to comment on this.

6) Page 10, lines 16-18. The statement is reasonable (although could be better worded); the zero order regime of O₂ is linked to conditions where the Cu oxidation state always remains unchanged, where there is a dependence on oxygen it is clearly linked to Cu oxidation state. This is a very nice finding. However, it strikes me that the authors should spend more time considering the potential for C₂H₄ to interact with Cu directly. This is commented on briefly with regards to CO₂ production, but not in great detail. Considering that C₂H₄ is known to interact with Cu(I) sites in zeolite Y (J. Phys. Chem. B 2004, 108, 17760-17766) this is a point that requires further consideration. By contemplating the 'side processes' and main reaction scheme holistically it may help to provide a more accurate reaction cycle.

7) Can the authors comment on the stoichiometry chosen for the low O₂ partial pressure. It appears as if it is sub-stoichiometric?

8) In Figure 4 (panels a,c,e) at time 20 minutes the C₂H₄ conversion plateaus, however the changes to Pd & Cu speciation continue to evolve. Can the authors comment on this? More broadly, it would be helpful if Figures 4 a and b can be adjusted to a scale that allows the readers to consider the carbon mass balance.

Answer to the reviewers' comments:

Reviewer 1:

1. Heterogenization of homogeneous Wacker Pd-Cu catalytic species may be an interesting subject, but the catalysis performance of the Pt-Cu/Y zeolite prepared in this study is low and the detailed characterization of this sort of catalytic materials would be almost meaningless in view of practical applications.

Authors: Characterization under non-steady state conditions necessitates the uniformity of chemical composition in the catalyst bed. As mentioned in our current manuscript (page 5, lines 19-21), operating at low ethylene conversions assumes differential reactor conditions, which minimize the concentration gradient from entrance to exit of the catalyst bed. Conversion is determined by the conditions such as residence time, temperature, pressure, amount of catalyst used, etc. A catalyst can have low conversion having high activity and vice versa. The activity of the catalyst (Pd1Cu5-Y) used in this study (*ca.* 5-10 mmol acetaldehyde · mmol Pd⁻¹ · min⁻¹) is one order of magnitude higher than those of the homogeneous systems (0.3-1.0 mmol acetaldehyde · mmol Pd⁻¹ · min⁻¹)^a. However, the homogeneous system is still widely utilized commercially because the catalyst solution can be regenerated continuously and efficiently.

The current study does not attempt to make the best material ripe for application since most of the work that helps understand catalytic mechanisms is done on systems that do not find application. Rather, the main aim of the current paper is to understand the mechanism of the heterogeneous Wacker process with the help of kinetic and transient spectroscopic studies. Transient studies provide mechanistic information that could not be extracted from steady-state conditions (as mentioned in the introduction, in page 4, lines 13-17). Ultimately, understanding the mechanisms of relevant industrial reactions has led to the engineering and development of more efficient catalysts.

[a] U.S. Patent, 3154586, 1964; U.S. Patent, 3149167, 1964; U.S. Patent 3301905, 1967.

2. "The conclusion on the behaviors and catalytic aspects of Pd(II)/Pd(0) and Cu(II)/Cu(I) is not new. The transient XAFS is also conventional, where the XAFS data for Pd and Cu have not been measured simultaneously but independently, and their Pt and Cu data appear to be combined on Figures."

Authors: We strongly disagree with the reviewer here. The dynamic behavior of the speciation of Pd and Cu under industrial Wacker conditions had never been reported elsewhere. The best attempt, with the techniques available to them at that time, was made by Espeel, et al. and is in detail discussed in this manuscript.

We did not claim that the transient XAFS we employed here is new but there is not even a single XAS study reported in the literature, be it under steady-state or transient conditions, for both heterogeneous and homogeneous Wacker processes. However, the state-of-the-art operando and transient quickXAS methods applied in this study is not "conventional" since the combination of these measurements with high time resolution and good data quality can only be done in specific beamlines at certain synchrotrons.

We did not state that Pd and Cu K-edge XAS spectra were measured simultaneously but their respective mass spectrometry response profiles (and gas feeding infrastructure) were similar so we

combined them in Figure 4. Nonetheless, the MS response profiles were only discussed qualitatively relative to the metal speciation.

3. “The discussion and conclusion on the local arrangements/structures of Cu(II) ions and Pd(II) ions argued from the XANES spectra are not convincing, where the changes in the XANES spectra for Cu and Pd species during the transient processes are very small though the authors made efforts to analyze by the MCR-ALS chemometric algorithm.”

Authors: A constructive comment on what in the discussion of the XANES spectra is not convincing would be very helpful. The separate qualitative discussion of the XANES spectra, before the chemometric (MCR-ALS) analysis, clearly discussed the specific changes in the XANES spectra that can be one to one related to speciation changes. The well-established MCR-ALS analysis of XAS data then allowed to extract the spectra of the purest components and showed the evolution of the presence of these components with time (as explained in the manuscript). This allowed for a quantitative description of the local structural re-arrangements in the system as a result of changing the kinetic regime. As indicated in the paper, the changes in the palladium fraction are small and gradual (page 11, lines 4-5). Whereas the transient experiments mostly affected the copper speciation without significant changes in the palladium fraction, demonstrating that oxygen is activated on copper.

4. “Other appropriate spectroscopic data for the coordination ligands should be presented for evidence.”

Authors: Action taken. We agree that more evidence would always be better but we exhaustively compared the XANES spectra of Pd1Cu5-Y to the standards available, indicating the presence of tetraammine palladium(II) complex and the hydrated copper(II) complex (Supplementary Figures 11 and 12), which were adequate for the discussion in the manuscript. Nonetheless, we added a complimentary IR-ATR characterization of the pre-treated Pd1Cu5-Y in Supplementary Fig. 5:

Supplementary Figure 5. FT-IR/ATR spectra of parent Na-Y and pretreated Pd1Cu5-Y

We added a sentence in the manuscript (page 7, lines 11-13): “This is further supported by the IR-ATR spectrum of Pd1Cu5-Y (Supplementary Fig. 5) exhibiting a weak band at 1310 cm^{-1} attributed to symmetric deformation of NH_3 coordinated to Pd(II).” We also discussed this further in the Supplementary Discussion (Supplementary Information, page 22, lines 1-6): “We also characterized the pre-treated catalyst with *ex situ* IR-ATR to confirm the presence of amminated Pd(II) (Supplementary Fig. 5). The weak band at 1310 cm^{-1} attributed to symmetric N-H deformation signifies the presence of amine coordinated to Pd(II). On the other hand, the absence of 1275 cm^{-1} band corresponding to the N-H deformation of $\text{Cu}(\text{NH}_3)_4^{2+}$ indicates that no ammonia was transferred from Pd(II) to Cu(II) after the ion exchange. The presence of a weak band at 1450 cm^{-1} indicates the release of some ammonia from Pd coordination.”

5. “Further, the responses of Cu(II)/Cu(I) and Pd(II)/Pd(I) in the transient processes of decreasing and increasing O₂ pressure were observed partially and the catalytic system always involved inactive Cu(II) and Pd (0) species heterogeneously in the catalyst. Thus, the current transient XAFS technique may not observe the real kinetic behavior of the active sites.”

Authors: We disagree with the reviewer, the presence of inactive Cu and Pd species does not mean that the transient processes were observed partially, but rather, that there are spectator species that do not participate in the reaction. This is the advantage of studying chemical reactions under transient conditions: we can distinguish between active and spectator species, which would be difficult to observe in steady-state conditions (as described in the introduction, page 4, lines 10-17): “The complexities in the interactions of the palladium and copper centers with the reactants and short life-times of intermediates during the Wacker process make it difficult to unambiguously identify the active species and the role, or lack thereof, of each component in the reaction mechanism under steady-state conditions. Therefore, we used transient XAS, with which the dynamic speciation of palladium and copper can be determined. When the kinetically relevant step is directly affected by disturbing the system, be it by changing the pressure, concentration or temperature, the active site yields a quick response while the spectator species would elicit a slower response²⁴.”

It is exactly the transient nature of the experiment that identifies what is active and what is not; species that do not respond to a change in oxygen environment and are active in a redox reaction are inactive; so the transient experiment actually highlights the active part of the catalyst.

6. In Figure 4 the quantity of the change in the acetaldehyde response does not agree with those of the changes in the Cu(II)/Cu(I) and Pd(II)/Pd(0) responses.

Authors: It is clearly indicated in the MS plot that the signals are normalized as we know that MS data without the proper corrections is not good enough to quantify the amount of reactants and products in the outlet. Nonetheless, we only used the MS response profiles in the discussion qualitatively, that is, to relate the relative changes in the m/z signals corresponding to each reactive gas to the changes in metal speciation.

7. The authors state that Figure 2 and Figure 4 are the transient results, but these are averaged aspects over 5 sec time resolution, and Figure 4 does not show elementary steps in Figure 5 for the catalysis. The authors describe the word “rapid”, “immediate”, etc. in several places,

but the time scales are 5 min or so. These slow responses may bring about the low active performance of the catalyst.

Authors: Action taken. The MS responses and change in metal speciation clearly show that there is no need to have a time resolution faster than 5 seconds which would have been extreme oversampling. It is evident in the time-resolved XANES spectra in Figure 2 and their respective MCR-ALS analysis (Figure 4a-d) that there are significant changes within tens of seconds, not the “5 min or so” as mentioned by the reviewer, which warrant the “rapid” and “immediate” descriptions as they are relative to all the changes in the MS response profiles and metal speciation. As mentioned in the introduction (page 4, lines 13-17), the rates by which palladium or copper change tell us what is active and what is not, not necessarily why the activity is low.

Thus, we changed the diction of some sentences and made our descriptions clearer:
(page 9, line 3): “**Within tens of seconds** after lowering the oxygen partial pressure,...”
(page 10, lines 23-25): “**Within tens of seconds** after time zero, a rapid increase in Cu(I) fraction from the reduction of Cu(II) is evident with a concomitant decrease in ethylene conversion.”
(page 11, lines 17-20): “After 60 minutes of low oxygen partial pressure (Fig. 4b,d,f), increasing the oxygen partial pressure back to the standard conditions led to a quick reoxidation of Cu(I) to Cu(II) **within tens of seconds** and equilibrated after 120 seconds, which corroborates the inference that oxygen is activated on copper.

And finally, we modified Fig. 2 slightly, to add and show more Cu K-edge XANES spectra in the beginning of the experiment (*ca.* 10-120 seconds after time zero).

Figure 2. Operando Cu and Pd K-edge XANES spectra of Pd₁Cu₅-Y under different reaction conditions. Normalized time-resolved (a,b) Cu and (c,d) Pd K-edge XANES spectra of Pd₁Cu₅-Y before (as-prepared, pre-treated, Wacker-equilibrated) and during transient experiments with (a,c) low pO_2 and (b,d) high pO_2 . The partial pressure of helium was adjusted to have a similar contact time all throughout the experiments. Inset: magnification of the (a) minor peak at 8983 eV corresponding to Cu(I) formation and (d) shift in absorption edge energy at half-height. Gas feed composition ($C_2H_4:H_2O:O_2$) at high pO_2 (1:7:10) and low pO_2 (1:7:0.05) was balanced with helium to achieve a constant contact time in all experiments

8. In the Mechanism section, one-electron transfers from Pd(0) to two Cu(II) ions, but the transient XAFS technique could not observe close locations to each other and no data for the synergistic effect of Cu and Pd on the Wacker catalysis were presented though the authors indicate their synergistic effect on the Wacker catalysis in Supplementary Figure 2.

Authors: Action taken. There is significant experimental evidence that the redox chemistry of the combined system differs from those with only either palladium or copper. We agree that EXAFS analysis reveal the presence of isolated copper and palladium ions and we repeatedly mentioned this in the manuscript (page 7, lines 23-25; page 13, lines 9-10). However, we also characterized the catalyst through other techniques. We have discussed in the manuscript that TPR (Supplementary Figure 2) indicates that their redox properties are coupled and that palladium and copper interact with each other and we therefore conclude that they are in close proximity. The XANES spectra (Supplementary Figures 11, 12) show the presence of solvated palladium and copper complexes, which are mobile and behave like ions in aqueous solutions. It is very likely that electron transport takes place by these mobile solvated species.

We also showed through STEM-EDX (see Figure below) that both palladium and copper are homogeneously distributed (within the resolution of the method) in the zeolite after ion exchange:

Supplementary Fig. 2-2. (a) HAADF-STEM image and the corresponding (b-e) EDX-STEM elemental maps of as-prepared Pd1Cu5-Y. Scale bar: 30 nm

Thus, we added these STEM-EDX images in Supplementary Fig. 2. We also added these sentences in the manuscript (page 5, lines 8-14): “HAADF-STEM and EDX elemental mapping of Pd1Cu5-Y (Supplementary Fig. 2) show that palladium and copper are homogeneously distributed in the zeolite after ion-exchange. Furthermore, the temperature-programmed reduction (TPR) profile of Pd1Cu5-Y (Supplementary Fig. 2) exhibits a single-peak feature indicating that their redox properties are coupled and that palladium and copper interact with each other. Hence, these results suggest that palladium and copper are in close proximity in Pd1Cu5-Y.”

Moreover, the synergistic effect was demonstrated as not having copper resulted in low activity while having no palladium resulted in zero activity, which qualifies as a synergistic effect because having both palladium and copper resulted in a much higher activity relative to just having either one of the them.

9. Finally, a proposed mechanism is shown in Figure 5, where unfortunately, the Pd sites as the Wacker reaction site have not been characterized and the argued mechanism is qualitative and even speculative with Cu sites as mentioned above.

Authors: The catalyst was characterized exhaustively with multiple techniques as discussed in the manuscript. Through meticulous XANES and EXAFS analyses, the electronic (oxidation state) and local atomic environment of both palladium and copper under operating conditions were investigated and the role of copper in performing redox chemistry beyond speculation was established. Utilizing chemometric algorithms allowed the quantitative analyses of palladium and copper speciation under transient conditions, which was then related to the MS responses of the reactive gases present qualitatively, ultimately leading to the proposed mechanism.

10. By careful overview and due to overall inconsistency of the approach and discussion, this manuscript has not provided new insight into the catalysis field, catalyst development and X-ray spectroscopy, and the reviewer cannot recommend this article for publication in Nature Communications.

Authors: We hope that our responses have made it clearer that the conclusions we have made were consistent to the results presented which provided these major breakthroughs:

- a. This is the first ever XAS study, under transient or steady-state conditions, on both homogeneous and heterogeneous Wacker processes; the applied methods are state-of-the-art, not “conventional”;
- b. We show how kinetic studies and transient XAS allowed us to identify the dynamic speciation of the catalyst in different reaction conditions with simultaneous activity measurements, providing structure-performance relationship information that has never been previously shown directly (qualitatively or quantitatively);
- c. First direct demonstration that oxygen is activated on copper in the heterogeneous system; we observed how Cu(I) reoxidation becomes rate-limiting at low oxygen partial pressure through the time resolution provided by quickXAS. The relation between catalyst speciation and a reactant order in the reaction mechanism is a seldom feature in the investigation of heterogeneous catalytic systems;
- d. The involvement of copper in the direct oxidation of ethylene to CO₂ which has not been mentioned or suggested previously.

Reviewer 2:

The paper 'Elucidating the mechanism of heterogeneous Wacker oxidation over Pd-Cu/zeolite Y by transient XAS' by Nachtegaal et al. describes the changes in Pd and Cu speciation, through operando XAS, as the oxygen partial pressure is varied for the heterogeneously catalysed Wacker oxidation. Overall, the paper is a well-executed piece of work of which the authors have clearly carried out with a great deal of care and rigour. This current paper focuses on the use of XAS to probe the mechanistic steps involved and corroborates many of the ideas put forward in the earlier work by Jacobs (J. Phys. Chem. 1994, 98, 11588-11596) performed using FTIR and EPR.

However, comparing the manuscript to the study by Jacobs I am a little unclear regarding what the major breakthrough is in this piece of work? The mechanism proposed has many similarities to the one put forward in that study and I am still unsure whether the data in the current manuscript wholly supports the proposed reaction cycle. To summarise, I think this is a very good study but the authors need to provide greater justification of the unique insight of this work.

Authors: We agree that the mechanism we proposed has similarities to the one proposed by Jacobs and coworkers but there were also differences, nonetheless. Espeel and coworkers monitored the electron transfer between Pd and Cu through *ex situ* ESR spectroscopy, which could not detect Cu(I) and Pd(II); and it was done under room temperature conditions, which deviate from industrial Wacker conditions. They also employed *in situ* IR spectroscopy under non-operational conditions (room temperature and low water partial pressure), and followed changes only every hour. They also used CO to probe the different redox states of Pd and Cu present but they found out that it was a non-neutral probe that reduced Pd(II), rendering their investigation indirect. Despite some similarities between our proposed mechanism and that of Espeel's, the characterization techniques we employed were different and allowed us to directly study the Wacker process and demonstrate these major breakthroughs in the current paper:

- a. It is the first ever XAS study, under transient or steady-state conditions, on both homogeneous and heterogeneous Wacker processes; the applied methods are state-of-the-art. (Page 4, lines 4-5): "However, there are no reported XAS studies focused on deciphering the active sites in heterogeneous or homogeneous Wacker oxidation."
- b. We show how kinetic studies and transient XAS allowed us to identify the dynamic speciation of the catalyst in different reaction conditions with simultaneous activity measurements, providing structure-performance relationship information that has never been previously shown directly (qualitatively or quantitatively)
- c. First direct demonstration that oxygen is activated on copper in the heterogeneous system; we observed how Cu(I) reoxidation becomes rate-limiting at low oxygen partial pressure through the time resolution provided by quickXAS. The relation between catalyst speciation and a reactant order in the reaction mechanism is a seldom feature in the investigation of heterogeneous catalytic systems. (page 4, lines 21-23): "Transient XAS at these different oxygen reaction orders showed a rapid redox reaction involving the Cu(II)/Cu(I) couple demonstrating directly for the first time that oxygen is activated on copper in the heterogeneous Wacker system."
- d. The involvement of copper in the direct oxidation of ethylene to CO₂ which has not been mentioned or suggested previously

Moreover, the reaction mechanism we propose highlights the changes observed in copper through the redox process involved. The transient experiments involved changes in oxygen partial pressure so it is expected that it would be the copper speciation that will be mostly affected following previously reported mechanisms for both homogeneous and heterogeneous Wacker processes. We modified the scheme (Figure 5) and the discussion (Page 13, lines 5-25; Page 14, lines 1-7) involving

the proposed mechanism to put emphasis on these results, highlighting the differences between our proposed mechanism and that of Espeel and coworkers'.

Figure 5. Proposed mechanism for heterogeneous Wacker oxidation of ethylene over Pd-Cu/Y. The catalytic cycle depicts the results obtained from this study. Reoxidation of palladium (*Step 2*) is represented by the dotted arrow. The activation of oxygen on copper (boxed) is depicted at high (solid arrow, *Step 3a*) and low (dashed arrow, *Step 3b*) oxygen partial pressures. The catalyst deactivation is shown in red.

My specific comments on the manuscript are as follows:

1) The comments on the advancement of synchrotron sources with respect to time resolution (page 4 lines 7-9) is correct in that things have improved with time – especially since the paper by Jacobs. However, measurements of this type have not been made possible by any recent advancement in the last 10 years. The authors may want to add a greater nuance to this part of the introduction.

Authors: Action taken. The sentence on page 4, lines 7-9, was rewritten: “The advent of high brilliance synchrotron sources, together with instrument developments, allows now to monitor structural changes occurring on sub-second to minute timescales, which would not have been possible a decade ago^{25,26}.” References [25] and [26] were added:

[25] Frahm, R. et al. The dedicated QEXAFS facility at the SLS: performance and scientific opportunities. *AIP Conf. Proc.* **1234**, 251 (2010).

[26] Fonda, E. et al. The SAMBA quick-EXAFS monochromator: XAS with edge jumping. *J. Synchrotron Rad.* **19**, 417-424 (2012).

2) The catalyst is prepared using a consecutive ion exchange process – 1st substituting with Cu and then Pd. The Cu substitution includes the details about the exchange process (concentration of solution, ratio to mass of zeolite, etc.), however, the same details are not given for the Pd step. It would be helpful if this can be included in the methods section alongside providing a justification for the sequential addition of metals and a comment on the total number of ion exchange sites (Na⁺) present per gram of zeolite. How does the PdCu loading introduced compare to the number of ion exchange sites?

Authors: Action taken. The sentence on page 14, lines 14-16 was rewritten: “Subsequently, the remaining sodium ions were exchanged in a 0.1 mM solution of Pd(NH₃)₄(CH₃COO)₂ (>99%, Alfa-Aesar) under similar conditions as the previous ion exchange with copper.”

The ion exchange with Pd(II) did not only remove sodium ions, but also some Cu(II) ions as there was a decrease in copper loading of the zeolite after ion exchange with Pd(II). Thus, there were still some sodium ions left after the two exchanges with copper and palladium.

3) Page 7 lines 22 & 23 – The comment about electron transfer/ion mobility is unsubstantiated and needs further justification.

Authors: Action taken. We agree that this needs further justification but we were clear to say that we only “suggest” that this ion mobility helps facilitate the electron transfer based on:

- a. Presence of solvated copper and palladium ions from their respective XANES spectra
- b. Evidence from the single peak feature from TPR of Pd1Cu5-Y indicating that palladium and copper are in close contact
- c. Evidence from STEM-EDX that palladium and copper are homogeneously distributed in the zeolite. We added the STEM-EDX images of Pd1Cu5-Y (Supplementary Figure 2-2) and a corresponding discussion in the manuscript (page 5, lines 8-14).
- d. Suggestions from previous cited studies (Page 8, lines 1-5) that solvated ions in wide-pored hydrophilic zeolite Y behave similarly to those in aqueous solutions.

Nonetheless, we changed and added these sentences in the manuscript to clarify the synergy and interaction between palladium and copper:

(page 7, line 25; page 8, lines 1-5): “However, its XANES spectra show the presence of solvated copper(II) and palladium(II) complexes in the wide-pored zeolite Y. As previously discussed through the results from STEM-EDX mapping and TPR studies, the close contact and electron transfer between the solvated copper and palladium complexes hint at their mobility within the zeolite, which have been suggested to have similar ionic character as that of palladium and copper ions in aqueous solutions^{8,15}.”

(page 13, lines 10-15): “However, the XANES spectra of the catalyst revealed that it is composed of mobile hydrated copper(II) and amminated palladium(II) ions. Moreover, H₂-TPR studies showed that the redox properties of palladium and copper are coupled while the STEM-EDX images of Pd1Cu5-Y illustrated their homogeneous distribution in the zeolite. These results suggest the interaction and proximity between palladium and copper, which are necessary for efficient electron-transfer.”

4) In general the figure captions do not contain sufficient detail, e.g., it would be helpful if the caption for figure 2 could include the ratio of all reactive gases (O₂, H₂O, C₂H₄) present in the feed. This is not really expressed clearly in the manuscript and instead needs to be extracted from the data points highlighted in figure 1. This is quite a useful piece of information to understand the manuscript and should also be mentioned in the main text before the figure is introduced.

Authors: Action taken. We completely agree that the ratio of all reactive gases should be stipulated and thus we added this sentence in the caption of Fig. 2 and Fig. 4: "Gas feed composition (C₂H₄:H₂O:O₂) at high p_{O_2} (1:7:10) and low p_{O_2} (1:7:0.05) was balanced with helium to achieve a constant contact time in all experiments."

5) The changes in the Pd K edge operando XANES study both qualitatively and through the multivariate analysis indicate the presence of Pd(0) increases between the 4h equilibrated form of the catalyst and at 1 h after switching to the low partial pressure of oxygen. There is no reason to suggest that this assessment is incorrect, however, the EXAFS does not show an increase in the Pd-Pd coordination number. It would be of value to comment on this in the manuscript alongside the values for the pseudo Debye-Waller factor. A value of 0.018 is larger than I would expect for 1st shell metallic Pd-Pd and it would be helpful for the authors to comment on this.

Authors: We agree that the Debye-Waller factor is unusually large for a 1st shell Pd-Pd. The changes in palladium during the transient experiments were also not that significant enough to elicit big changes in the Pd-Pd coordination number, which would be well within the standard error. The relatively large pseudo Debye-Waller factor could point to the large disorder from the presence of both particles and single atoms.

We added a sentence in the manuscript (page 8, lines 16-17): "However, the relatively large pseudo Debye-Waller factor of the metallic Pd-Pd shell suggests the large disorder from the presence of both particles and single atoms."

6) Page 10, lines 16-18. The statement is reasonable (although could be better worded); the zero order regime of O₂ is linked to conditions where the Cu oxidation state always remains unchanged, where there is a dependence on oxygen it is clearly linked to Cu oxidation state. This is a very nice finding. However, it strikes me that the authors should spend more time considering the potential for C₂H₄ to interact with Cu directly. This is commented on briefly with regards to CO₂ production, but not in great detail. Considering that C₂H₄ is known to interact with Cu(I) sites in zeolite Y (J. Phys. Chem. B 2004, 108, 17760-17766) this is a point that requires further consideration. By contemplating the 'side processes' and main reaction scheme holistically it may help to provide a more accurate reaction cycle.

Authors: Action taken. Thank you very much for this information. We added a sentence in the manuscript to address this (page 11, lines 11-12): Although ethylene is known to adsorb on Cu(I)³⁸, this interaction is insufficient to lead to full oxidation at low oxygen partial pressures.

Ref. [38] was added: Datka, J. & Kukulska-Zajac, E. IR studies of the activation of C=C bond in alkenes by Cu⁺ ions in zeolites. *J. Phys. Chem. B* **108**, 17760-17766 (2004).

7) Can the authors comment on the stoichiometry chosen for the low O₂ partial pressure. It appears as if it is sub-stoichiometric?

Authors: Action taken. The amount of oxygen is sub-stoichiometric with respect to ethylene. The low O₂ partial pressure in the transient experiments was chosen based on the results of the kinetic studies. Moreover, spectroscopically, it elicited the most significant changes in the time-resolved XAS.

We added the sentence in the Methods Section in the Manuscript (page 15, lines 17-19): “The sub-stoichiometric amount of oxygen at low partial pressure was chosen from the kinetic studies eliciting the largest changes in the time-resolved XAS spectra.”

8) In Figure 4 (panels a,c,e) at time 20 minutes the C₂H₄ conversion plateaus, however the changes to Pd & Cu speciation continue to evolve. Can the authors comment on this? More broadly, it would be helpful if Figures 4 a and b can be adjusted to a scale that allows the readers to consider the carbon mass balance.

Authors: No action taken. We agree that the changes in the m/z signals for ethylene equilibrated starting at around 25 min, albeit with a slight increase. Nonetheless, the MS responses were a bit noisy considering the low amounts of C₂H₄ and its eventual conversion to acetaldehyde. However, the copper speciation loosely follows that trend with slight changes after 25 min. This amount of copper at 25 min (Figure 4c) is close to the value at t<0 before the start of the switch to high oxygen partial pressure (Figure 4d).

On the other hand, the MCR-ALS results for Pd speciation fluctuates more (larger error) near the end of the measurement from 30-45 min after t=0 in the first switch to low oxygen partial pressure (Figure 4e). The value of Pd fraction at 25 min (considering this larger error) also only gradually and slightly changed until the 45th minute, and is close to the value at t<0 before the switch to high partial pressure (Figure 4f).

As previously mentioned (response to comment #4), we were cautious to extract quantitative information from the MS response profiles so we did not calculate the carbon balance. However, based on the plots we have shown (Supplementary Figure 3) illustrating the activity of the catalyst under standard conditions, the carbon balance of the acetaldehyde and carbon dioxide products is close to 1, albeit always less than 1. This could mean two things:

- a. There are side-products that were not characterized, most likely acetic acid based on previous studies. However, acetic acid was not detected in GC and MS based on separate measurements.
- b. Some of the products and side-products were adsorbed onto the catalyst leading to the less-than-1 carbon balance. Supplementary Table 2 shows that the carbon content of the spent catalyst after 4 hours is 5.4%, corroborating the inference that there are carbon-containing deposits on the catalyst.

REVIEWERS' COMMENTS:

Reviewer #1 (Remarks to the Author):

The current transient XAFS technique is useful and the authors have done elaborate experiments and analysis. Nevertheless, the reviewer cannot see what progress in catalysis research has been made by the finding by this technique and the reviewer does not think that the current work has provided new insight into the origin and uniqueness of the catalysis mechanism for the heterogeneous PdCu/Y catalyst.

Reviewer #2 (Remarks to the Author):

The authors have diligently responded to the concerns raised in my initial review and that of the additional reviewer.

As I stated in my first review, this is a 'well-executed piece of work of which the authors have clearly carried out with

a great deal of care and rigour'. My major concern was the level of unique insight. Here, the authors have certainly strengthened their manuscript and I am happy to support publication.

On the more minor points raised, the authors have taken these comments seriously and provided an adequate response. I have only one comment on these points, where the authors may wish to revise their text. The authors commented:

We added a sentence in the manuscript (page 8, lines 16-17): "However, the relatively large pseudo Debye-Waller factor of the metallic Pd-Pd shell suggests the large disorder from the presence of both particles and single atoms."

It is unclear to me how a single atom (by definition devoid of Pd neighbours) adds to the disorder of Pd-Pd interactions.

Answer to the reviewer's comments:

Reviewer 2:

The authors have diligently responded to the concerns raised in my initial review and that of the additional reviewer.

As I stated in my first review, this is a 'well-executed piece of work of which the authors have clearly carried out with a great deal of care and rigour'. My major concern was the level of unique insight. Here, the authors have certainly strengthened their manuscript and I am happy to support publication.

On the more minor points raised, the authors have taken these comments seriously and provided an adequate response. I have only one comment on these points, where the authors may wish to revise their text. The authors commented:

We added a sentence in the manuscript (page 8, lines 16-17): "However, the relatively large pseudo Debye-Waller factor of the metallic Pd-Pd shell suggests the large disorder from the presence of both particles and single atoms."

It is unclear to me how a single atom (by definition devoid of Pd neighbours) adds to the disorder of Pd-Pd interactions.

Authors: Action taken. The sentence was rewritten: "The relatively large pseudo Debye-Waller factor of the metallic Pd-Pd shell suggests a large disorder due to the presence of Pd clusters: increasing Debye-Waller factors are associated with decreasing particle size³³." Reference (34) was added:

(34) Frenkel, A., Hills, C. & Nuzzo, R. A view from the inside: Complexity in the atomic scale ordering of supported metal nanoparticles. *J. Phys. Chem. B* **105**, 12689-12703 (2001).